# ONC201 and an MEK Inhibitor Trametinib Synergistically Inhibit the Growth of Triple-Negative Breast Cancer Cells

**DOI:** 10.3390/biomedicines9101410

**Published:** 2021-10-07

**Authors:** Bora Lim, Christine B. Peterson, Alexander Davis, Elin Cho, Troy Pearson, Huey Liu, Minha Hwang, Naoto Tada Ueno, Jangsoon Lee

**Affiliations:** 1Section of Translational Breast Cancer Research, Department of Breast Medical Oncology, Morgan Welch Inflammatory Breast Cancer Research Program and Clinic, The University of Texas MD Anderson Cancer Center, Houston, TX 77030, USA; tpearson@mdanderson.org (T.P.); huey.liu@mdanderson.org (H.L.); cherish01290@gmail.com (M.H.); nueno@mdanderson.org (N.T.U.); 2Breast Oncology, Baylor College of Medicine, Houston, TX 77030, USA; 3Department of Biostatistics, The University of Texas MD Anderson Cancer Center, Houston, TX 77030, USA; CBPeterson@mdanderson.org; 4Department of Genetics, The University of Texas MD Anderson Cancer Center, Houston, TX 77030, USA; AJDavis2@mdanderson.org; 5The University of Texas Health Science Center at Houston, Houston, TX 77030, USA; sungjin.cho@uth.tmc.edu

**Keywords:** TNBC, ONC201, MEK inhibitor, apoptosis, trametinib

## Abstract

Triple-negative breast cancer (TNBC) is a heterogeneous group of estrogen, progesterone, and HER2-negative breast cancers with poor clinical outcomes. The imipridone ONC201 is a G-protein-coupled dopamine receptor D2 modulator and an allosteric agonist of the mitochondrial protease caseinolytic protease P(ClpP), which induces apoptosis. Here, we aimed to develop a novel ONC201-based combination therapy targeting TNBC. We performed a reverse-phase protein array analysis of ONC201-treated/-untreated and -sensitive/-resistant cell lines to identify potential predictive biomarkers. A principal component analysis using measured protein expression levels, the apoptosis score (AS), and heatmaps of all the measured protein and AS-related protein expression levels did not show a clear correlation between the expression levels of a specific protein and ONC201 efficacy. Three-dimensional RNA interference kinome-wide library screening revealed the MAPK and PI3K/Akt pathways as potential synergistic therapeutic partners. The combination with the MEK inhibitor trametinib successfully inhibited the growth of both ONC201-sensitive/-resistant TNBC cell lines. The baseline ClpP level correlated with the efficacy of single-agent ONC201. Single and combination therapy increased caspase 3/7 activity. The predictive biomarkers and a detailed mechanism of synergy beyond an induction of caspase activation should be tested for translation into future studies.

## 1. Introduction

Although new targeted therapeutics, such as sacituzumab govitecan-hziy (anti-TROP2 antibody-drug conjugate) [1] and immunotherapeutics, have been effective against triple-negative breast cancer (TNBC) [2], patients still suffer from therapy resistance and disease progression. The evasion of apoptosis is a critical mechanism of therapy resistance of cancers, even more so of cancers with TP53 alterations. About 83% of TNBCs harbor TP53 mutations or functional TP53 loss due to the loss of heterozygosity [3]. Thus, we hypothesized that inducing apoptosis would be an essential therapeutic strategy for TNBC. Indeed, researchers are actively developing mitochondrial apoptosis-inducing therapeutics for breast cancers. Balko et al. [4] reported that MCL1 was amplified in about 58% of residual cancers after neoadjuvant chemotherapy in patients with early-stage TNBC. An MCL1 inhibitor is currently being developed in preclinical settings [5,6]. Recently, the BCL-2 inhibitor venetoclax exhibited excellent efficacy when combined with endocrine therapy for estrogen receptor-positive breast cancers [7] and entered testing for the treatment of HER2-positive breast cancers overall and TNBCs in particular. Inhibitor of apoptosis protein inhibitors and other apoptosis modulators also produced promising results in both preclinical and clinical studies [8].

ONC201, a small molecule imipridone, is a modulator of the G-protein-coupled dopamine receptor D2 and an allosteric agonist of the mitochondrial protease caseinolytic protease P (ClpP), inducing apoptosis in multiple solid tumors [9,10]. It also induces a G-protein-coupled receptor-mediated tumor necrosis factor-related apoptosis-inducing ligand activity and subsequent apoptosis in a ClpP-dependent manner [11]. Thus, ONC201 is an attractive potential therapeutic for TNBC. However, most small molecules do not show robust efficacy when given as single agents. Therefore, we hypothesized that ONC201 would induce the death of TNBC cells when combined with other targeted small molecules. To test this, we screened such a synergistic small molecule inhibitor partner and confirmed the synergistic efficacy of the discovered inhibitors using in vitro and ex vivo models. We also examined the ClpP expression level and its correlation with ONC201 sensitivity in TNBCs.

## 2. Material and Methods

### 2.1. TNBC Cell Lines and the Half-Maximal Inhibitory Concentration of ONC201

BT-20, HCC38, HCC70, HCC1187, HCC1395, HCC1806, HCC1937, MDA-MB-157, MDA-MB-231, MDA-MB-453, and MDA-MB-468 cells were obtained from the ATCC (Manassas, VA, USA). SUM149, SUM159, and SUM185 cells were obtained from Asterand Bioscience (Detroit, MI, USA). HCC2185 and HCC3153 cells were purchased from The University of Texas Southwestern Medical Center (Dallas, TX, USA). CAL51 and CAL120 cells were obtained from the Leibniz Institute *DSMZ* (Braunschweig, Germany). All cell lines were validated using short tandem repeat DNA profiling at The University of Texas MD Anderson Cancer Center Cytogenetics and Cell Authentication Core and confirmed to be free of mycoplasma infection. Moreover, all cell lines were maintained according to the suppliers’ guidelines, tested for mycobacterial contamination, and screened to determine the range of half-maximal inhibitory concentrations (IC_50_s) of ONC201. Trametinib, VX-11e, MK-2206, PF0491052, buparlisib, and dactolisib were purchased from Selleck Chemicals (Houston, TX, USA). Pre-validated ClpP siRNA were purchased from Sigma-Aldrich (St. Louis, MO, USA). Sequences are 5′-GCUCAAGAAGCAGCUCUAU-3′, 5′-CGCUCAUUCCCAUCGUGGU-3′, 5′-CCAUGGAGAGGGACCGCUA-3′. Each cell line was treated with ONC201 alone at different dose levels and analyzed using a CellTiter-Blue cell viability assay (Promega, Madison, WI, USA) and sulforhodamine B assay to assess tumor-growth inhibition according to the manufacturer’s instructions. The synergy between ONC201 and other tested drugs was analyzed using an isobologram and the combination index (CI) with CalcuSyn software (v2.1; BIOSOFT, Cambridge, UK).

### 2.2. Three-Dimensional RNA Interference Kinome-Wide Library Screening

To identify the genes that can enhance the antitumor efficacy of ONC201, RNA interference (RNAi) screening of a library of 779 kinomes and 160 G-protein-coupled receptors was performed under three-dimensional (3D) growth conditions using ONC201-sensitive CAL51 and ONC201-resistant HCC70 TNBC cells. Reverse transfection of cell lines with the siGENOME Kinome siRNA Library (Horizon Discovery, Lafayette, CO, USA) was used to identify partner of inhibition that changes the cell growth inhibition of ONC201. In brief, 20 µL of a small interfering RNA (siRNA) solution (200 nM of a pool of four siRNA duplexes in Opti-MEM medium) was mixed with 20 µL of DharmaFECT 1 (0.24 µL in Opti-MEM medium; Invitrogen, Carlsbad, CA, USA) in a 96-well NanoCulture plate (MBL International, Woburn, MA, USA). After 20 min of incubation at room temperature, 1 × 10^4^ TNBC cells were added to the NanoCulture plate. Forty-eight hours after reverse transfection, cells were treated with ONC201 at the 30% inhibitory concentration to examine the synergistic tumor-growth inhibition. After 72 h of incubation, cell viability was determined using CellTiter-Glo 3D Cell Viability Assay reagent (Promega) according to the manufacturer’s instructions. Details on this method were described by our group previously [12].

### 2.3. Reverse-Phase Protein Array

Reverse-phase protein array (RPPA) analysis was performed using two ONC201-sensitive cell lines (MDA-MB-468 and CAL51) and two ONC201-resistant cell lines (MDA-MB-157 and SUM159) as described previously [13]. ONC201-treated and un-treated cell lines were submitted to this analysis in triplicate. Briefly, proteins extracted from cultured cells were denatured and serially diluted to define antigen-antibody reactions in a linear range for accurate quantification using nitrocellulose-coated slides (Grace Bio-Labs, Bend, OR, USA) and tyramide dye deposition and a diaminobenzidine colorimetric reaction (CSA reaction kit; DAKO, Glostrup, Denmark). Quantitative analysis was performed using software featuring background correction, controlling for location, and concentration determination (Array-Pro and SuperCurve fitting). The RPPA were optimized to measure the levels of 300 proteins for downstream analysis. Normalized log2-transformed values of data were used throughout the reported analysis [14].

### 2.4. Apoptosis Score Calculation, Principal Component Analysis, and Hierarchical Clustering

Single-sample gene set enrichment analysis (GSEA) [15] was conducted to calculate the apoptosis score (AS), including a total of 24 apoptosis-related genes. An enrichment score for each sample was calculated, reflecting how genes in a prespecified list rank in terms of abundance in the sample relative to that in the other samples. The AS was computed using the GSVA package in the R computing language [16]. Principal component analysis (PCA) was performed to measure the differences in the protein levels among the cell lines and the therapeutic effect of ONC201.

### 2.5. Three-Way Analysis of Variance of the RPPA Data to Determine ONC201′s Therapeutic Effect

Three-way analysis of variance was conducted using the R language to determine whether our analysis model fits significantly better by including a dependence of ONC201′s treatment effect on resistance. To calculate *p*-values, the null model was set as protein level = cell line average + treatment effect, without including resistance. The alternative hypothesis is that the treatment effect is different in the resistant cells by an amount “Δ resistant”. For example, if a cell line is sensitive to treatment with ONC201, the protein level goes up due to “treatment effect” only. However, if the cell line is resistant to this treatment, the protein level changes by “treatment effect + Δ resistant” to separate the resistance from the direct therapeutic effect. This teased out the potential predictive synergistic partners of ONC201.

Formally, the model for each protein level is
$$X_{jk}=\mu_{j}+G\times I{\left(\mathrm{treated}\right)}_{k}+\Delta \times I{\left(\mathrm{treated}\;\mathrm{and}\;\mathrm{resistant}\right)}_{jk}+\epsilon_{jk}$$
where $X_{jk}$ is the level of the protein in cell line j with treatment k (where k is either ONC201 or control), $\mu_{j}$ is the mean protein level for this cell line without treatment, G is a parameter representing the effect of treatment, $I{\left(\mathrm{treated}\right)}_{k}$ is 1 if the treatment k was ONC201 and 0 if it was control, Δ is a parameter representing the change in treatment effect in a resistant cell line, $I{\left(\mathrm{treated}\;\mathrm{and}\;\mathrm{resistant}\right)}_{jk}$ is 1 if the cell line j is one of the two ONC201-resistant cell lines described above and the treatment was ONC201, and 0 otherwise, and $\epsilon_{jk}$ is normally distributed noise. The null hypothesis is that Δ=0. *p*-values were adjusted using the Bonferroni-Hochberg correction [17].

### 2.6. Ex Vivo Tumor-Growth Inhibition Assay

To form 3D cell spheroids using xenograft tissue samples, NanoShuttle-PL and a 96-well bioprinting kit (Greiner Bio-One, Monroe, NC, USA) were used according to the manufacturer’s instructions. In brief, 4- to 6-week-old female NOD/SCID gamma mice (Jackson Laboratory, Bar Harbor, ME, USA) were housed under specific pathogen-free conditions and cared for following National Institutes of Health guidelines. Animal studies were approved by the MD Anderson Institutional Animal Care and Use Committee (00000968-RN02). HCC70 and CAL51 TNBC cells were collected and resuspended in a 50% Matrigel solution at 5 × 10^7^ cells/mL. Tumor samples were collected when the tumor volume was about 400 mm^3^ and dissociated using a gentleMACS automatic tissue dissociator (Miltenyl Biotec, Auburn, CA, USA) according to the manufacturer’s instructions. Collected tumor cells were treated with red blood cell lysis buffer (11814389001; Sigma, St. Louis, MO, USA) and then, using dead cell removal kits (130-090-101; Miltenyl Biotec). Cells (5 × 10^4^), were resuspended in 200 µL of high-glucose Dulbecco’s modified Eagle’s medium (10% fetal bovine serum) with 5 µL of NanoShuttle-PL, added to a low-attachment 96-well plate (3474; Corning, Corning, NY, USA), and incubated on a magnetic plate for 2 days to drive spheroid formation. After 5 days incubation, the diameter and viability of the spheroids were measured using an Eclipse Ti microscope (Nikon Instruments Inc., Melville, NY, USA) and CellTiter-Glo 3D cell viability assay reagent (G9683; Promega), respectively.

### 2.7. Western Blot Assay

A Western blot assay using a Wes Simple Western instrument (ProteinSimple, San Jose, CA, USA) was performed according to the manufacturer’s instructions. In brief, whole-cell extracts from TNBC cell lines were prepared using M-PER mammalian protein extraction reagent (Thermo Fisher Scientific, Waltham, MA, USA), and the total protein concentration was quantified using Pierce BCA reagents (Thermo Fisher Scientific). Lysates of TNBC cell lines were diluted to 0.2 µg/µL in M-PER reagent and denatured using Fluorescent Master Mix (ProteinSimple) at 95 °C for 5 min. Anti-ClpP (1:50 dilution; Sigma-Aldrich, St. Louis, MO, USA), anti-SDHB (1:50 dilution; Abcam, Cambridge, MA, USA), and anti-α-tubulin (1:300 dilution, T5168; Sigma-Aldrich) antibodies were used as primary antibodies, and an Anti-Rabbit Detection Kit (DM-001; ProteinSimple) or Anti-Mouse Detection Kit (DM-002; ProteinSimple) was used for secondary antibodies according to the manufacturer’s instructions.

### 2.8. Caspase 3/7 Assay

To quantify cellular apoptosis after therapy with ONC201, the MEK inhibitor trametinib, a Caspase-Glo 3/7 assay (G8090; Promega), or both were performed according to the manufacturer’s instructions. In brief, TNBC cells were seeded into a 96-well plate at 1 × 10^4^ cells/well and incubated overnight. Cells were then treated with ONC201 (2.5 µM), trametinib (1 µM), or both for 24 h. Equal volumes of Caspase-Glo 3/7 assay buffer were added to the wells with cells and incubated for 1 h at 37 °C. The luminescent intensity of each well was measured using a Victor X3 microplate reader (PerkinElmer, Waltham, MA, USA).

### 2.9. Statistical Analysis

For in vitro experiments, descriptive statistics (mean and SD) were summarized for each group. Statistical analyses of two groups were performed using an unpaired *t*-test with Prism software (v6; GraphPad Software, San Diego, CA, USA). For RPPA analysis, three-way analysis of variance was used with the R program package. *p*-values less than 0.05 were considered significant. The CI and fraction affected were determined using CalcuSyn (v2.1) to evaluate the synergistic effect of ONC201 in combination with other drugs.

## 3. Results

### 3.1. The IC_50_ of ONC201 Varies among TNBC Cell Lines

We first measured the anti-proliferation efficacy of ONC201 in 17 TNBC cell lines. We observed a dose-dependent anti-proliferation effect in the tested cell lines by ONC201 treatment (Appendix A), and the IC50s range is from 2.05 to 43.39 µM (Table 1). To define the ONC201-sensitive and non-sensitive TNBC cell line, we referred to the Greer et al. report that the ONC201 IC50 in solid tumors sensitive to this agent was about 5 µM [9]. Thus, we classified the TNBC cell lines as sensitive or resistant to treatment with ONC201 based on these data. Next, we investigated whether ONC201 IC50 is correlated with the original Vanderbilt TNBC molecular subtype, a transcriptomic subtyping that was developed by Pitenpol’s group with an aim to categorize the heterogeneous TNBC into therapeutically targetable subgroups [18]. We did not observe an association of the TNBC subtypes with ONC201 sensitivity (Appendix A).

### 3.2. The 3D RNAi Kinome Library Screening Identified MAPK and PI3K/Akt Inhibitors as Potential Synergistic Partners of ONC201

Next, we performed 3D RNAi kinome library screening to identify potential kinase targets to enhance the antitumor effect of ONC201 in TNBC cells. We selected the ONC201-sensitive cell line CAL51 (2.05 μM) and ONC201-resistant cell line HCC70 (12.06 μM) for the screening. We identified 233 genes in CAL51 (Appendix A) and 279 genes in HCC70 (Appendix A) as potential partners that would enhance the therapeutic efficacy of ONC201 in TNBC. We found that 65 genes in the two target gene sets overlapped (Appendix A). Next, we performed an Ingenuity Pathway Analysis of these 65 genes to identify the relevant canonical pathways for combination with ONC201. Five canonical pathways—NFAT regulation of immune response, interleukin-8 signaling, Gq signaling, PTEN signaling, and ephrin receptor signaling—were relevant pathways (Figure 1A). We then ran a STRING protein interaction assay to identify key target proteins and detected PIK3CA, MAP4K4, and AKT3 as potential target proteins (Figure 1B). Based on this result, we selected MEK, PI3K, PI3K/mTOR, and Akt inhibitors for testing as potential synergistic partners of ONC201 in TNBC treatment.

### 3.3. PCA and Clustering of RPPA-Based Protein Levels Did Not Show Robust Correlation between Measured AS and ONC201 Sensitivity

To examine the protein expression variation among TNBC cells with different levels of sensitivity to treatment with ONC201, we performed a principal component analysis of the full set of 300 measured protein expression levels from the RPPA. We subjected two ONC201-sensitive cell lines (CAL51 and MDA-MB-468) and two ONC201-resistant cell lines (SUM159 and MDA-MB-157) to either ONC201 treatment or no treatment in triplicate to measure the protein levels in the RPPA. The PCA showed a difference among the TNBC cell lines regardless of ONC201 sensitivity. PC1 captured the largest proportion of data variance (24.7%) and showed the separation of the protein expression levels among the cell lines, confirming the diversity of the protein expression levels among these lines. Differences in the levels of proteins between the treated and untreated cell lines were also evident, as shown in PC2, which captured an additional 19.7% of the overall data variance, except for a somewhat weaker separation between treated and untreated MDA-MB-157. The within-group variance for the triplicates’ value of protein levels under the same treatment conditions and in the same cell lines was relatively low by comparison (Figure 2A). We computed an Apoptotic Score (AS) using 24 apoptosis-relevant proteins to measure a comprehensive grouped level of apoptotic activity. Based on this protein list, we calculated the AS for each sample via single-sample GSEA using 24 protein levels (Appendix A). Overall, the ASs in the ONC201-resistant TNBC cell lines before and after the treatment with ONC201 were more similar to each other, whereas those in the ONC201-sensitive cells were more different. The ONC201-sensitive cell lines had significantly higher baseline ASs than did the ONC201-resistant cell lines. However, they became similar to those in the resistant cell lines after the ONC201-based treatment (Figure 2B).

Next, we created heat maps of all the protein and AS-related protein expression levels using unsupervised hierarchical clustering of the protein expression levels from each sample based on their similarity across the full set of 300 proteins (Figure 2C). Next, we performed hierarchical clustering using only the proteins used to compute the AS. Among the 24 proteins, the level of MCL1 increased after the treatment with ONC201, with a higher degree of changes noted in the ONC201-sensitive cell lines than in the ONC201-resistant cell lines. The level of PARP protein expression in the ONC201-sensitive cell lines decreased substantially after the ONC201-based treatment. The rest of the proteins in this analysis did not exhibit significant changes in protein levels between the ONC201-sensitive and -resistant cell lines in any direction. When we compared the untreated and ONC201-treated cells as groups, we found that the levels of phosphorylated S6 proteins differed considerably in the ONC201-sensitive and -resistant cell lines (Figure 2D).

### 3.4. Protein Levels and Their Correlation with ONC201′s Therapeutic Effects

Because the PCA plot and hierarchical clustering of the RPPA data demonstrated that both TNBC cell lines and treatment status make a substantial contribution to the variation to the level of protein as independent contributing factors, we performed a three-way analysis of variance that included individual TNBC cells’ characteristics, acknowledging that unique cell characteristics affect protein expression levels. We used an adjusted *p*-value less than 0.05 and a coefficient greater than 1 (plus and minus) for this analysis. Defining the “treatment effect” on a given protein as the difference between the protein expression in the ONC201-treated and untreated cells of the same cell line, we identified seven proteins where the treatment effect in the resistant cell lines was significantly different than in the sensitive cell lines. These proteins did not directly overlap with the genes discovered in the RNAi kinome library screening. High EMA, HER2_pY1248, PLK1, and Rb pS807/811 protein expression had treatment effects that were more positive in the resistant cell lines. Thus, inhibiting these targets may synergize with ONC201 in targeting TNBC. In contrast, SOD2, PAR, and fibronectin protein expression displayed more negative treatment effects in the resistant cell lines (Table 2).

### 3.5. MEK Inhibitor Trametinib Enhances the Antiproliferative Effect of ONC201 in TNBC Cells

3D RNAi kinome library screening revealed that the PI3K/AKT/mTOR and MAPK pathways are potential targets for potentiating the antitumor effect of ONC201. To validate these findings, we performed a combination assay using seven targeted therapy partners—the MAPK inhibitors trametinib (MEKi), ulixertinib (ERKi), and VX-11e (ERKi) and the PI3K/Akt/mTOR pathway inhibitors MK-2206 (AKTi), PF04691052 (AKTi and mTORi), buparlisib (PI3Ki), and dactolisib (PI3Ki)—and the TNBC cell lines MDA-MB-453, MDA-MB-231, SUM149, and HCC70. We observed that trametinib (CI, 0.07–0.95), ulixertinib (CI, 0.08–1.09), VX-11e (CI, 0.17–0.97), and MK-2206 (CI, 0.1–1.23) exhibited synergism with ONC201 but that PF04691052 (CI, 0.33–3.07), buparlisib (CI, 0.28–3.98), and dactolisib (CI, >1.0) did not (Table 3, Appendix A). The fractional effect and combination index are presented in Appendix A.

Next, we examined the synergistic antitumor effects of ONC201 and the MEK inhibitor trametinib. We conducted an ex vivo treatment of the ONC201-sensitive (CAL51) and -resistant (HCC70) TNBC xenograft tissue samples with ONC201 with and without trametinib. We first evaluated the antitumor effect of ONC201 alone. We observed that ONC201 showed a dose-dependent growth-inhibitory effect in both CAL51 and HCC70 tissue samples (*p* < 0.0001) (Figure 3A). Next, we evaluated the therapeutic effect of the combination of ONC201 and trametinib. We found that the combination more significantly reduced the growth of tumor spheroids than did treatment with ONC201 or trametinib alone (*p* < 0.001, Figure 3B) and synergistically reduced the viability (ONC201, 64% in CAL51 and 44% in HCC70; trametinib, 90% in CAL51 and 32% in HCC70; combination, 99% in CAL51, 96% in HCC70) of both CAL51 and HCC70 ex vivo models (*p* < 0.0001) (Figure 3C).

### 3.6. ONC201 Sensitivity Is Dependent on the ClpP Expression Level in TNBCs and Induces Caspase 3/7 Activity in Combination with Trametinib

The detailed mechanism of action has been reported by Ishizawa et al. [19]. In this study, ONC201 activated the protease and facilitated the activation of ClpP and the ClpP mediated cleavage of substrates within the mitochondria. The induction of mitochondrial ClpP activation leads to the reduction of the respiratory chain complex subunit SDHB level [19]. To further analyze the mechanism of the therapeutic action of ONC201 in our models, we focused on a known mechanism, the ClpP-mediated inhibition of mitochondrial respiratory chain complex subunits. We compared the levels of SDHB and ClpP expression in different treatment groups using a Western blot assay. We observed that ONC201 significantly reduced the ClpP and SDHB expression when administered alone and in combination with trametinib in both ONC201-sensitive (CAL51) and -resistant (HCC70) TNBC cell lines (Figure 4A). ONC201 alone and with trametinib also reduced the ClpP expression. However, trametinib alone did not. We next investigated the median levels of ClpP expression in TNBC cell lines and found that the IC_50_ of ONC201 correlated with ClpP expression (*p* = 0.0446) (Figure 4B). We then explored whether ClpP is a crucial molecule in the ONC201-mediated antitumor effect by inducing the overexpression of ClpP using an expression vector and downregulating ClpP using RNAi (Appendix A). We found that ClpP-overexpressing TNBC cells responded to ONC201-based treatment (Figure 4C), whereas ClpP-downregulated TNBC cells did not (Figure 4D). We also confirmed that treatment with trametinib did not regulate the ClpP expression (Appendix A).

To determine whether TNBC cells had undergone apoptosis by the combination treatment with ONC201 and trametinib, we tested the activity of caspase 3 and 7 in TNBC cells treated with a vehicle (control), ONC201 alone, trametinib alone, or ONC201 and trametinib. In ONC201-sensitive CAL51 cells, the caspase 3/7 activity increased with the single-agent of ONC201 (1.75-fold), trametinib (3.13-fold), and combination treatments (6.6-fold). The differences in the effect on caspase 3/7 activity between therapy with ONC201 alone and the combination (*p* < 0.0001) and between that with trametinib alone and the combination (*p* < 0.05) were significant (Figure 4E). In ONC201-resistant HCC70 cells, the caspase activity increased with single-agent therapy with both ONC201 (1.33-fold) and trametinib (1.30-fold) to the same degree. The combination therapy significantly increased the activity of caspase 3/7 (1.88-fold, *p* < 0.001) (Figure 4E).

## 4. Discussion

ONC201 is a new drug with a superb safety profile in normal cells tested in the treatment of multiple cancers, including ovarian and breast cancers. Given its safety profile in normal cells and that it penetrates the central nervous system, ONC201 has high translational potential. The present study is the first to demonstrate the therapeutic efficacy of ONC201 in combination with trametinib in TNBC cell lines. We confirmed that the expression of a known direct target of ONC201, ClpP, correlates well with ONC201′s single-agent efficacy, suggesting that other potential predictive biomarkers and synergistic partners of ONC201 should be tested.

RNAi kinome library screening identified the inhibitors of the MAPK and PI3K/Akt pathways as potential synergistic partners of ONC201. ONC201′s known mechanism of action is directly inducing an unfolded protein response by mitochondrial restructuring to induce apoptosis. Interestingly, these two pathways are upstream regulators of apoptosis induction via other mechanisms. Thus, we hypothesized that the inhibitors of these two pathways can synergistically enhance the ONC201 efficacy. In addition to identifying partners of ONC201, we sought to identify predictive biomarkers of ONC201′s efficacy in TNBC treatment by analyzing the RPPA data. Whereas we confirmed that ONC201 induced caspase 3/7 activity in both ONC201-sensitive and -resistant TNBC cell lines, the AS calculated using 24 apoptosis-regulating proteins correlated with the ONC201 sensitivity as a total score. In addition, we identified several proteins that correlated with ONC201 sensitivity regardless of the unique TNBC cell characteristics in our three-way analysis of variance. The two proteins correlating with ONC201 sensitivity with the lowest expression levels were fibronectin (a glycoprotein that recruits extra cellular matrix and cytoskeleton scaffolding proteins via integrin, e.g., laminin, vinculin, paxillin and α-actinin.), PAR (a Cytoplasmic scaffolding proteins), and G-protein-coupled receptors. ONC201 directly binds to the mitochondrial protein ClpP to cause structural changes and a subsequent stress response. Thus, these scaffolding proteins may be important to ONC201′s efficacy in TNBC treatment. Another protein, SOD2, is a predictive marker of the sensitivity of TNBC to treatment with ONC201 in that ONC201 induces reactive oxygen species production. Thus, a high level of SOD2 expression may induce the therapeutic efficacy of ONC201.

Expression increases in four proteins—HER2_pY1248, PLK1, Rb_pS807/811, and EMA—have been correlated with resistance to ONC201. For example, HER2_pY1248 is a critical catalytic site of the HER2 receptor. The intact cell-cycle regulator Cdks phosphorylates Rb activity, and the proteins that regulate the spindle and centromere function, EMA and PLK1, are also correlated with ONC201 sensitivity. These findings suggested the importance of the complex mechanisms of ONC201 activity against TNBC that can be examined in future clinical studies.

We next confirmed the MEK inhibitor trametinib as the new therapeutic combination partner of ONC201, among the potential synergistic partners discovered from the RNAi library screening. The synergistic efficacy of ONC201 and trametinib was evident when tested in an ex vivo assay and in both ONC201-sensitive (CAL51) and ONC201-resistant (HCC70) TNBC cells. Moreover, as shown previously in other cancers (Jo’s paper), the ONC201 sensitivity of TNBC correlated with the level of ClpP expression. However, treatment with trametinib did not affect the level of ClpP expression, confirming the hypothesis that the synergy is not through the further reduction of ClpP protein expression. Rather, we found that the mechanism of synergy of ONC201 and trametinib occurs through the enhanced induction of caspase activity. The combination treatment with ONC201 and trametinib increased caspase 3/7 activity in TNBC cells, confirming mitochondrial apoptosis activation by this therapy.

However, our study has a limitation of not clearly teasing out a detailed mechanism of synergy between the trametinib and ONC201 beyond the induction of caspase 3, 7 mediated apoptosis. Moreover, in our limited scope of this study, we did not validate the potential predictive biomarker of ONC201 anti-tumor efficacy. Future studies to address these areas would further strengthen the translation of this novel combination we have identified.

## 5. Conclusions

We confirmed our hypothesis that the treatment with ONC201 in combination with the MEK inhibitor trametinib synergistically inhibits the growth of TNBC cells regardless of ONC201′s activity alone. The ClpP expression level in TNBC cells at the baseline correlated with ONC201 sensitivity, which could be rescued by the administration of siRNA ClpP, yet the combination of ONC201 and trametinib did not reduce the expression of ClpP further. Instead, the combination increased caspase 3/7 activity. In addition to the correlation between the AS and ONC201 sensitivity of TNBC, we discovered a correlation between the resistance and more positive treatment effect on EMA, HER2_pY1248, pRb sS807, and PLK1 and the resistance and more negative treatment effect on PAR, fibronectin, and SOD2 by analyzing four TNBC cell lines using an RPPA. These potential resistance mechanisms should be tested further, which could strengthen the translational potential of our identified novel combination therapy in TNBC in future clinical studies.

## Figures and Tables

**Figure 1 biomedicines-09-01410-f001:**
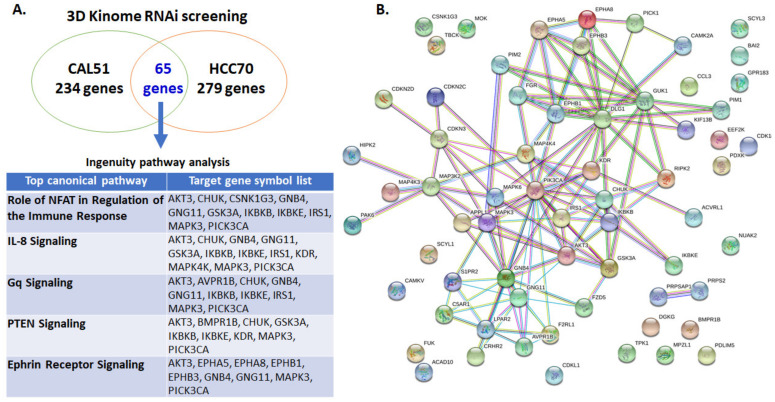
3D kinome siRNA library screening using the TNBC cell lines CAL51 (ONC201-sensitive) and HCC70 (ONC201-resistant) identified 65 overlapping genes in the two cell lines that synergistically suppress the growth of the cells with ONC201. (**A**) Ingenuity Pathway Analysis of these genes showing five major pathways that were relevant canonical pathways: NFAT regulation of immune response, interleukin (IL)-8 signaling, PTEN signaling, Gq signaling, and ephrin receptor signaling. A STRING protein interaction assay further identified *PIK3CA*, *MAP4K4*, and *AKT3* as potential therapeutic target genes. (**B**) The STRING analysis of 65 overlapping target genes.

**Figure 2 biomedicines-09-01410-f002:**
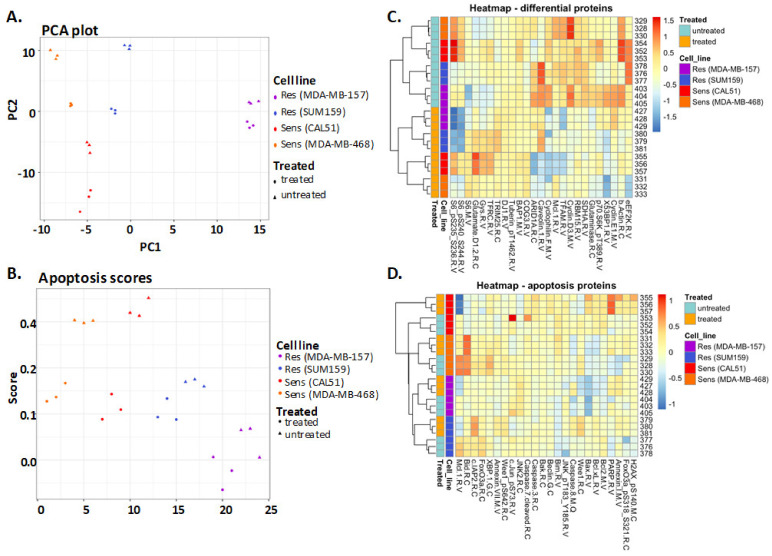
Gene set-exclusive analysis using apoptotic score (AS) gene sets in association with ONC201-sensitive versus -resistant TNBC cell lines using RPPA data. (**A**) PCA results showing that the degree of overall 24 apoptosis-related protein expression changes before and after ONC201-based treatment was much greater in ONC201-sensitive (Sens) cell lines but smaller in ONC201-resistant (Res) cell lines. PC1, averaged protein level of each cell line; PC2, average protein level of each treatment condition. (**B**) Results of PCA according to AS in four TNBC cell lines showing differences in AS score induced by ONC201-based treatment in the cell lines that were ONC201-sensitive and -resistant. The differences were not significant. (**C**,**D**) Heat maps of the (**C**) differential and (**D**) apoptosis protein levels in each cell line according to an RPPA assay, which do not show proteins significantly correlated with ONC201 sensitivity.

**Figure 3 biomedicines-09-01410-f003:**
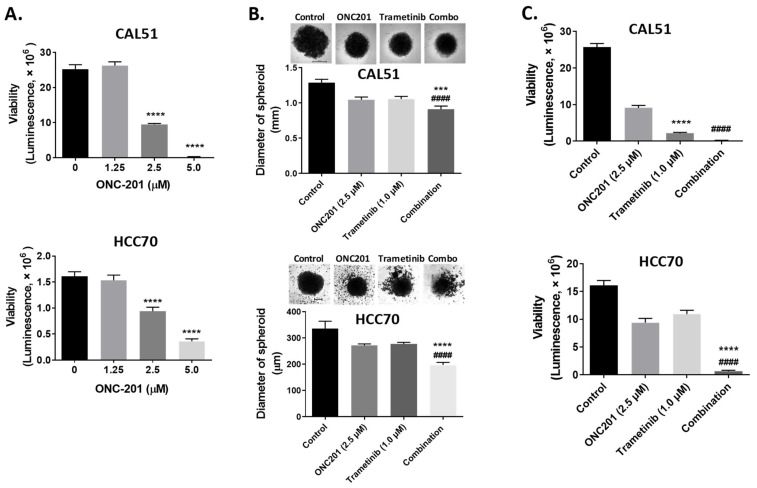
Results of an ex vivo combination study of treatment with ONC201 and the MEK inhibitor trametinib using the CAL51 (ONC201-sensitive in vitro) and HCC70 (ONC201-resistant in vitro) TNBC cell lines. (**A**) Graphs of the viability of CAL51 and HCC70, which was significantly reduced by single-agent treatment with ONC201. An ONC201 concentration of 2.5 µM was selected as the combination therapy dose. (**B**) Spheroids size of CAL51 and HCC70 tumor. (**C**) Spheroid viability after treatment with ONC201 (2.5 µM) and trametinib (1 µM) showing similar tumor-growth inhibition when compared with the control treatment. Combination treatment with ONC201 and trametinib synergistically reduced the 3D growth of both CAL51 and HCC70 spheroids more so than did single-agent treatment with them. *** *p* < 0.001, **** *p* < 0.0001. #### *p* < 0.0001.

**Figure 4 biomedicines-09-01410-f004:**
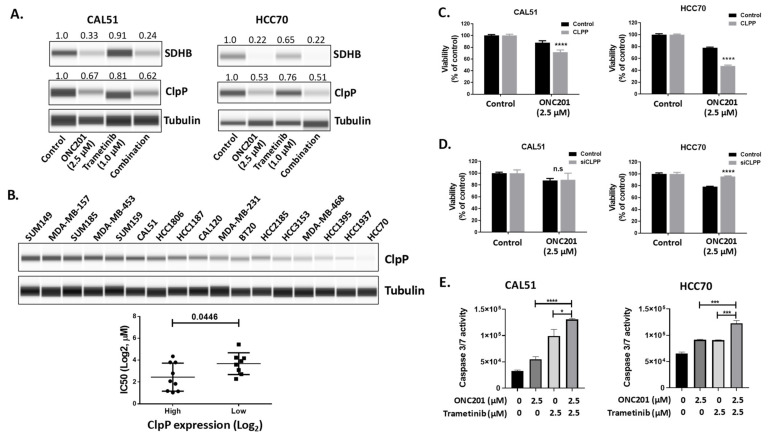
Assessment of the known direct targets of ONC201, SDHB, and ClpP in TNBC cell lines. Cells were treated with DMSO control, ONC201 alone (2.5 µM), trametinib alone (1 µM), or a combination of ONC201 and trametinib. (**A**) Western blots showing that ClpP and SDHB levels were markedly reduced by ONC201 in both ONC201-sensitive (CAL51) and -resistant (HCC70) TNBC cell lines. (**B**) Western blot data showing that the median level of ClpP expression was significantly correlated IC_50_ of ONC201 in TNBC cell lines (*p* = 0.0446). (**C**,**D**) The cells transfected with a ClpP expression vector or siRNA for 48 h and then treated with ONC201 for 5 days, and then cell viability was measured by sulforhodamine B assay. (**E**) Graphs showing that treatment with ONC201 in combination with trametinib induced caspase 3/7 activity in CAL51 and HCC70 cells. Cells were treated with ONC201 (2.5 µM) with or without trametinib (1 µM) for 24 h, and a caspase 3/7 activity assay was performed. n.s, not significant, * *p* < 0.05; *** *p* < 0.001; **** *p* < 0.0001 (unpaired Student *t*-test).

**Table 1 biomedicines-09-01410-t001:** The IC_50_s of ONC201 in TNBC cell lines according to subtype (2011 Vanderbilt classification). TNBC cells had varying degrees of sensitivity to ONC201. We did not observe any correlations of TNBC subtype with ONC201 IC_50_. BL1: Basal-like 1, BL2: Basal-like 2, M: Mesenchymal, LAR: Luminal androgen receptor.

Subtype	Cell Line	IC_50_ (µM)
BL1	HCC1937	18.73
	MDA-MB-468	4.86
	HCC3153	15.11
BL2	HCC70	12.06
	HCC1806	6.57
	SUM149	2.26
M	CAL51	2.05
	CAL120	4.22
	MDA-MB-157	13.94
	MDA-MB-231	6.57
	SUM159	20.36
LAR	HCC2185	43.39
	SUM185	13.92
	MDA-MB-453	3.58
Other	BT20	8.54
	HCC1395	18.10
	HCC1187	2.22

**Table 2 biomedicines-09-01410-t002:** Protein levels and their correlation with ONC201′s therapeutic effects.

Protein	*p*-Value	Coefficient	Adjusted *p*-Value
EMA	1.340 × 10^−2^	4.464	3.489 × 10^−2^
Fibronectin	1.994 × 10^−3^	−1.203	8.687 × 10^−3^
HER2_pY1248	3.130 × 10^−3^	1.420	1.182 × 10^−2^
PAR	1.385 × 10^−4^	−2.204	1.113 × 10^−3^
PLK1	6.535 × 10^−6^	1.690	8.970 × 10^−5^
Rb pS807/811	1.994 × 10^−3^	1.644	8.687 × 10^−3^
SOD2	3.143 × 10^−4^	−1.078	2.019 × 10^−3^

EMA (MUC1): Epithelial membrane antigen, PAR: poly(ADP-ribose, PLK1: polo-like kinase 1, SOD2: Superoxide dismutase 2.

**Table 3 biomedicines-09-01410-t003:** Synergistic antitumor effect of ONC201 in combination with MAPK and PI3K pathway inhibitors.

TNBC Cell Line	Pathway Inhibitors						
	Trametinib (MEKi)	Ulixertinib (ERKi)	VX-11e (ERKi)	MK-2206 (AKTi)	PF04691052 (PI3K/mTORi)	Buparlisib (PI3Ki)	Dactolisib (PI3Ki)
MDA-MB-453	S	N	S	S	N	N	N
MDA-MB-231	S	S	S	S	N	N	N
SUM149	S	S	S	S	S	S	N
HCC70	S	S	S	S	N	N	N

Abbreviations: MEKi, MEK inhibitor; ERKi, ERK inhibitor; AKTi, AKT inhibitor; PI3K/mTORi, PI3K/mTOR inhibitor; PI3Ki, PI3K inhibitor; S, synergism with ONC201; N, no synergism with ONC201.

## Data Availability

The dataset(s) supporting the conclusions of this article is(are) included within the article (and its Appendix A).

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
