# Peer review of "ONC201 and an MEK Inhibitor Trametinib Synergistically Inhibit the Growth of Triple-Negative Breast Cancer Cells"

_biomedicines, 2021, doi:10.3390/biomedicines9101410_

Round 1
Reviewer 1 Report
The study entitled “ONC201 and an MEK inhibitor trametinib synergistically inhibit the growth of triple-negative breast cancer cells” demonstrates that the inhibitors of the MAPK and PI3K/Akt pathways could be potential synergistic partners of ONC201 and shows the ability of MEK inhibitor trametinib to increase the efficacy of ONC201 in TNBC cells. Although the subject of the manuscript is interesting the authors should address some points to make the manuscript suitable for publication.
- The files of supplementary figures and tables have not been loaded
- The authors have to report in the text the same IC50s range reported in Table 1 from 2.05 to 43.39 mM
- The authors have to add the abbreviations of TNBC subtypes under Table 1
- The authors should add in the introduction the Vanderbilt classification and discuss it
- The authors have to add the abbreviations of TNBC subtypes under Table 2
- I suggest to authors report CI for each inhibitor in Table 3
- In figure 3B and C the same unit should be reported in the graphs for the two cell lines
- Authors need to better discuss why ONC201 treatment reduces SDHD and CIpP levels and what the underlying mechanism might be.
- The symbols in the text have to be corrected
- Although the authors used TNBC xenografts, they did not evaluate the effect of drugs on tumor growth in vivo, Why?
Author Response
Reviewer 1.
- The files of supplementary figures and tables have not been loaded
We did not realize that the supplemental data were not correctly uploaded. Hopefully you can see the data now.
- The authors have to report in the text the same IC50s range reported in Table 1 from 2.05 to 43.39 mM
We apology for typos. IC50 range is from 2.05 to 43.39 µM. We have corrected the information in the table and manuscript to be consistent.
- The authors have to add the abbreviations of TNBC subtypes under Table 1
We have added the abbreviations of TNBC subtype in the Table 1.
- The authors should add in the introduction the Vanderbilt classification and discuss it
We have added a brief description in the section where the Vanderbilt subtype is introduced I line 166-167).
- The authors have to add the abbreviations of TNBC subtypes under Table 2
We have added the abbreviations of EMA. PAR, PLK1, SOD2 in the Table 2.
- I suggest to authors report CI for each inhibitor in Table 3
We apology for missing supplementary data. CI values provided in Supplemental table 5.
- In figure 3B and C the same unit should be reported in the graphs for the two cell lines
We have updated drug concentration and unit in all figures.
- Authors need to better discuss why ONC201 treatment reduces SDHD and CIpP levels and what the underlying mechanism might be.
Thankfully this mechanism has been studied by numerous investigators, and deconvoluted by Ishizawa et al. in 2019 ( PMID: 31056398). We have added brief description into the result section line between 256-259. Hopefully we have addressed the concern of the reviewer.
- The symbols in the text have to be corrected
We apology for typos. We carefully reviewed the manuscript and corrected all symbols.
- Although the authors used TNBC xenografts, they did not evaluate the effect of drugs on tumor growth in vivo, Why?
We also recognized that xenograft experiment is critical to validate our findings. However, unfortunately, we do not have enough research fund to conduct xenograft experiment. Therefore, we decided to utilize ex vivo technology as an alternative approach which widely use for drug development. Our group have shown that ex vivo result showed similar drug response compared to xenograft models. (PLoS One. 2018 May 16;13(5):e0195932. doi: 10.1371/journal.pone.0195932. PMID: 29768500)

Reviewer 2 Report
Lim et al. use an integrative approach of experimental and computational techniques to access the synergetic effect of drugs against triple-negative breast cancer cells. Overall, the paper points towards interesting resistance mechanisms, it is very well written, methods are carefully exposed, detailed statistics are made in every step, and conclusions are supported by data.I only want to draw the authors' attention to a few points: Fig 2. Quality should be improved, especially at panels A and B as it is very difficult to perceive the differences. Hierarchical clustering specific details are not listed in the methodology section Is there a specific reason to use a variety of mathematical software’s for the different analyses instead of just R? Table 2 – The number of significant figures seems unnecessarily high.Author Response
Reviewer 2.
- Fig 2. Quality should be improved, especially at panels A and B as it is very difficult to perceive the differences.
We apology for the low quality of images. We placed with high quality images.
- Hierarchical clustering specific details are not listed in the methodology section Is there a specific reason to use a variety of mathematical software’s for the different analyses instead of just R?
Indeed, the analysis was performed using R packages, or R languages. The three way analysis we tried to perform here did not fit the exact language within the GSVA package, therefore we wanted to lay out how we programmed the R language. If this is confusing, we can move the section to the supplemental information. Please let us know about this.
- Table 2 – The number of significant figures seems unnecessarily high.
Thank you for the comment. We reduced the numbers of significance.
